# Comparative Transcriptome Analysis Provides Novel Insights into the Effect of Lipid Metabolism on Laying of Geese

**DOI:** 10.3390/ani12141775

**Published:** 2022-07-11

**Authors:** Qingyuan Ouyang, Shenqiang Hu, Bincheng Tang, Bo Hu, Jiwei Hu, Hua He, Liang Li, Jiwen Wang

**Affiliations:** Farm Animal Genetic Resources Exploration and Innovation Key Laboratory of Sichuan Province, Sichuan Agricultural University, Chengdu 611130, China; oyqy222@163.com (Q.O.); shenqiang.hu@sicau.edu.cn (S.H.); i_cbtang@163.com (B.T.); hubolirong@163.com (B.H.); hujiwei1990@126.com (J.H.); hehua023@126.com (H.H.); liliang@sicau.edu.cn (L.L.)

**Keywords:** lipid metabolism, transcriptome, egg production, goose

## Abstract

**Simple Summary:**

The importance of lipid metabolism in the egg production of poultry has been widely reported. Meanwhile, geese have lower egg production and unique lipid metabolism patterns compared with chicken and duck. It is of great significance to further improve egg laying performance to explore the differences of fat metabolism and the molecular mechanisms in geese with different egg laying performance. This study compared the phenotypic differences of liver and abdominal fat, as well as the transcriptome level differences of liver, abdominal fat, and ovarian stroma among high-, low-, and no-egg production groups. The results reveal that lipid metabolism regulated by the circadian rhythm of the liver may directly or indirectly affect ovarian function through the inflammation and hormone secretion of abdominal fat.

**Abstract:**

The lower egg production of geese (20~60 eggs per year) compared with chicken and duck limits the development of the industry, while the yolk weight and fatty liver susceptibility of geese was higher than that of other poultry. Therefore, the relationship between lipid metabolism and the laying performance of geese remains to be explored. Phenotypically, we observed that the liver fat content of the high-, low-, and no-egg production groups decreased in turn, while the abdominal fat weight increased in turn. For transcriptional regulation, the KEGG pathways related to lipid metabolism were enriched in all pairwise comparisons of abdominal fat and liver through functional analysis. However, some KEGG pathways related to inflammation and the circadian rhythm pathway were enriched by DEGs only in abdominal fat and the liver, respectively. The DEGs in ovarian stroma among different groups enriched some KEGG pathways related to ovarian steroidogenesis and cell adhesion. Our research reveals that lipid metabolism regulated by the circadian rhythm of the liver may directly or indirectly affect ovarian function through the inflammation and hormone secretion of abdominal fat. These results offer new insights into the regulation mechanisms of goose reproductive traits.

## 1. Introduction

Compared with chicken and duck, the lower egg production of geese (20~60 eggs per year) limits the development of the industry. With the development of cage breeding technology of geese, we found that different individuals have different abilities to lay eggs, and there are even individuals that never lay eggs. Previous studies showed that lipids gradually accumulate from small white, small yellow to large yellow during the follicle development stage of geese [1,2]. Meanwhile, the weight of goose egg yolk is much greater than that of chicken and duck, and it may take more time to deposit enough yolk. Therefore, it is speculated that lipid metabolism (synthesis, transport, deposition) may be one of the important factors affecting the egg production of geese.

It has been proven that the ovaries of geese have the ability of de novo lipid synthesis [3] Meanwhile, most of the lipids in follicles (more than 90%) come from the transport of liver [4]. Compared with chicken, geese have a stronger capacity for lipid metabolism in the liver [5]. Geese are commonly used to produce fatty liver, and some scholars have pointed out that the difference in lipid metabolism caused by overfeeding is more obvious in the liver than in adipose tissue. The liver involves pathways such as lipid metabolism and immune response, while the adipose tissue involves pathways such as protein binding and gluconeogenesis after overfeeding. Part of the lipids synthesized by the liver are transported to the ovary to participate in the formation of egg yolk, and the others are stored in the form of sebum or abdominal fat. For obese individuals with a large amount of fat deposited in the abdominal cavity, fat will compress the ovary and fallopian tube, reducing the function of the reproductive system and affecting the formation of eggs and ovulation rate, resulting in a decline in egg production [6]. Too much or too little fat deposition has a negative effect on the initiation of egg production [7].

Studies have shown that the increase in egg production of broilers under feed restriction is due to the different amounts of lipid transport from the liver to the ovaries or sebum/abdominal fat [8]. It has been reported that lipid metabolism differences in ovarian stroma are involved in regulating egg production in geese [9]. Therefore, this study aims to explore the pathways of different lipid metabolisms affecting the reproductive performance of geese by constructing the mRNA expression profiles of liver, abdominal fat, and ovarian stroma of geese among the no-, low-, and high-egg production geese. These data will provide new insights into the regulation mechanism of goose reproductive traits.

## 2. Materials and Methods

### 2.1. Sample Collection

A total of 232 female Sichuan white geese hatched in the same batch were reared in single cages after the end of the brood period. The laying performance of 232 geese was recorded until 365 days of age. The geese that no eggs laid were divided into the no egg production group (*n* = 40), the geese with the top 30% egg production were divided into the high egg production group (*n* = 54), and the geese with the bottom 30% of egg production were divided into the low egg production group (*n* = 54). Meanwhile, we weighed 232 geese. Then, five individuals with similar body weight and consistent physiological states were selected from the three groups for slaughter. Abdominal fat and liver weight were recorded. Livers were dissected and placed onto a tissue mold, filled with an optimal cutting temperature compound (Fisher Scientific, Pittsburgh, PA, USA), and frozen at −20 °C. Next, abdominal fat, the partial liver, and ovarian stroma were collected and placed in liquid nitrogen.

### 2.2. Histological Observation

Each liver tissue was taken to a size of 24 × 24 × 2 mm, frozen and placed on a tissue-supporting device dripping with OTC embedding agent. Tissue blocks were frozen and leveled. The tissue was cut into 10 m thick slices using a CM1520 slicer (Leica, Weztlar, Germany) and attached to the anti-slip slide. Distilled water was washed for 2 min, 60% isopropanol aqueous solution (Keshi, Shanghai, China) for 2 s, oil red O working staining solution (Sigma, Shanghai, China) for 15 min, 60% isopropanol aqueous solution for 2 s, running water for 2 min, and hematoxylin (Thermo Fisher Scientific, Waltham, MA, USA) for 30 s. Then, neutral resin adhesive was used to seal the film. The staining of tissue samples was observed under a microscope (OLMPUS, Tokyo, Japan). The microscopic imaging system was used to take photos and record the staining results. Image Pro Plus 6.0 software (Media Cybernetics, Inc., Rockville, MD, USA) was used to analyze the average optical density of the positive results of oil red staining.

### 2.3. RNA-Seq and Bioinformatics Analysis

Liver, abdominal fat, and ovarian stroma were selected from 3 individuals in each group. The Trizol kit (Invitrogen, Santa Clara, CA, USA) was used to extract the total RNA of liver, abdominal fat, and ovarian stroma according to the manufacturer’s instructions. The RNA integrity was determined by an Agilent Bioanalyzer 2100 (Agilent Technologies, Santa Clara, CA, USA). The RNA samples were used for library construction. The mRNA libraries were sequenced by Novogene Co., Ltd. (Beijing, China) using Nova-PE150 (Illumina, San Diego, CA, USA). The clean reads were obtained after the filtration of low-quality reads using standard quality control by FastaQC software. Clean reads were mapped to the Anser cygnoides domestication reference genome (data being published) using the HISAT2 (version 2.2.1) software [10]. The output SAM (sequencing alignment/mapping) file was converted to a BAM (binary alignment/mapping) file and sorted using SAMtools (version 1.10) [11]. Subsequently, the expression of each transcript was calculated by featureCounts (version 1.6.0) [12]. DEseq2 was used to identify the different expression genes (DEGs) among different groups, the screening criteria were |log_2_Foldchange| > 1, *p*-_adjust_ < 0.05. Functional analysis used KOBAS 3.0 online (http://kobas.cbi.pku.edu.cn/kobas3/?t=1, accessed on 26 August 2021) [13].

### 2.4. Quantitative Real-Time PCR Validation

Total RNA extracted from the tissues was reverse transcribed into cDNA using a Taq Pro Universal SYBR qPCR Master Mix (Vazyme Co., Ltd., Nanjing, China). Primer 5.0 was used to design the primers (Table 1). A BLAST search against the reference genome was then carried out to confirm that primers were specific for the intended target genes. Taq Pro Universal SYBR qPCR Master Mix (Vazyme, Nanjing, China) and a Bio-Rad CFX96 real-time PCR detection system (Bio-Rad, Hercules, CA, USA) were used for RT-PCR, and each sample was assayed three times. *β-actin* and *GAPDH* were used as housekeeping genes. The 2^−∆∆CT^ method was used for normalization of the qPCR results, after which the normalized data were used for statistical analysis, and *p* < 0.05 was considered significantly different.

### 2.5. Statistical Analysis

The abdominal fat weight/index, liver weight/index, fat count of liver, and expression levels of DEGs were expressed as the mean ± standard deviation (SD). One-way ANOVA was used for statistical analysis in this study. *p* < 0.05 was considered statistically significant. All statistical analyses were carried out using the SPSS 27.0 software.

## 3. Results

### 3.1. The Lipid Deposition Patterns of Geese with Different Laying Performance

There was no significant difference in body weight among the three groups (Figure 1A). However, the abdominal fat weight and abdominal fat index of geese in the low- and no-egg production groups were significantly (*p* < 0.05) higher than those in the high-egg production group (Figure 1B). In contrast to the weight of abdominal fat, the content of lipids in the liver of the high-egg production group was significantly (*p* < 0.05) higher than that of the low- and no-egg production groups.

### 3.2. Overview of the mRNA Transcriptome with Different Egg Production Performance Geese

A total of 672,265,178 raw reads were obtained from 27 samples through mRNA sequencing, and 90.65% of the clean reads were aligned to the goose reference genome (Appendix A). As shown in Figure 2A, the number of DEGs in ovarian stroma with different egg performances was lower than that of liver and abdominal fat. Meanwhile, the number of unique DEGs between the no- and high-egg production group was the largest in the three tissues (Figure 2B–D).

### 3.3. Functional Analysis of DEGs among Different Egg Production Performances in Abdominal Fat, Liver, and Ovarian Stroma

The DEGs in almost groups were significantly enriched in several KEGG pathways (Appendix A). Of note, the KEGG pathways related to lipid metabolism (e.g., cholesterol metabolism, fat digestion and absorption, bile secretion) were enriched in all pairwise comparisons of abdominal fat and liver (Figure 3). However, some KEGG pathways related to inflammation (IL-17 signaling pathway, NF-kappa B signaling pathway, inflammatory mediator regulation of TRP channels, leukocyte transendothelial migration, Toll-like receptor signaling pathway, inflammatory bowel disease) and the circadian rhythm pathway were enriched by DEGs only in abdominal fat and liver, respectively. Meanwhile, the hormone-related KEGG pathways (thyroid hormone synthesis, GnRH signaling pathway, insulin signaling pathway, and estrogen signaling pathway) were enriched by DEGs in abdominal fat. The DEGs in ovarian stroma enriched some KEGG pathways related to ovarian steroidogenesis (ovarian steroidogenesis and oxytocin signaling pathway) and cell adhesion (focal adhesion, ECM-receptor interaction, and adherens junction).

### 3.4. Network Construction of Liver, Abdominal Fat, and Ovarian Stroma Regulating Laying Performance in Geese

We chose key DEGs in liver (enriched in lipid metabolism and circadian rhythm), abdominal fat (enriched in inflammatory response, lipid metabolism, and hormone), and ovarian stroma (enriched in steroid hormone and cell adhesion) to construct a protein–protein interaction network (PPI) (Figure 4A). It was found that the DEGs in liver, abdominal fat, and ovarian stroma have a mutual regulatory relationship through the PPI. DEGs with more than 10 nodes were used for qPCR verification in this network (Figure 4B). As shown in Figure 4B, expression of almost all these selected DEGs displayed changes in the same direction with those observed using RNA-seq, indicating the true reliability of our sequencing and analysis methods.

## 4. Discussion

Phenotypically, we found that liver lipid content and abdominal fat weight had opposite trends in the high-, low-, and no-egg production groups. Meanwhile, a previous study of broilers showed a positive relationship between liver lipid content and abdominal fat weight [14,15] This difference between chicken and geese may be due to the unique lipid metabolism pattern of geese (domesticated by migratory birds and high lipid storage capacity of liver) [16]. The abdominal fat weight of the no-egg production group was higher than that of the other groups. Consistent with that, the previous results showed that excessive abdominal fat delays the start of production in poultry [17], and that excessive abdominal fat reduces the egg production performance of female poultry [18]. Our results also confirm this point.

Through the transcriptome results, we found that the number of DEGs in liver and abdominal fat was greater than that in ovarian stroma. These results showed that energy metabolism organs play a vital role in female poultry reproduction activities [19,20]. The liver and fat are the most important tissues of lipid metabolism [21,22]. Meanwhile, we found that the KEGG pathways related to lipid metabolism were enriched in all pairwise comparisons of abdominal fat and liver by the functional analysis of DEGs, while the circadian rhythm pathway was enriched by DEGs only in the liver. Circadian rhythms are physiologic and behavioral cycles that control a variety of biological processes, including feeding, the sleep–wake cycle, and the female reproductive cycle [23]. Many studies have shown that circadian rhythms in the ovary is important for reproduction [24,25]. However, our study first showed that the circadian rhythms in liver is related in poultry reproduction. The dynamic balance of liver energy depends on enzymes, and these rate-limiting enzymes are expressed in a circadian rhythm [26,27]. In view of the circadian rhythm of the liver in mammals, Reinke et al. reviewed its functions in carbohydrate metabolism, lipid metabolism, amino acid metabolism, detoxification, synthesis of plasma proteins, and bile acid metabolism [28]. The important role of *NPAS2* [29,30], *RORG* [31], and *PER3* [32] in the circadian regulation of lipid metabolism has been repeatedly demonstrated. Studies in mammals have shown that *NPAS2* not only activates the expression of *PER* but is also the target gene of *ROR* [33]. In our study, *NPAS2*, *RORG*, and *PER3* were differentially expressed in the liver of geese among different laying groups. Hence, we speculate that the difference of circadian rhythm in the liver may cause the differences of lipid metabolism.

Meanwhile, we found that some KEGG pathways related to inflammation, including the IL-17 signaling pathway, NF-kappa B signaling pathway, inflammatory mediator regulation of TRP channels, leukocyte transendothelial migration, Toll-like receptor signaling pathway, and inflammatory bowel disease KEGG pathways, were enriched by DEGs in abdominal fat. The positive relationship between obesity and inflammation has been widely established [34,35]. In our results, multiple inflammation-related genes (*IL8* and *CCL4*) showed a trend of low to high expression in the abdominal fat of the high-, low-, and no-egg production groups. The negative effects of inflammation are multifaceted, including for reproduction [36]. Adipose tissue has been considered as an endocrine organ capable of secreting hormones that travel through the bloodstream to reach their target tissues [37]. In addition, the DEGs in abdominal fat are also enriched in the hormone-related KEGG pathways. Therefore, the change in hormone-related gene expression in abdominal fat may directly affect the laying performance of geese.

The ovarian stroma is directly related to follicle development and laying, and DEGs in ovarian stroma are enriched in the KEGG pathways associated with steroid hormones and cell adhesion. Lipids are not only important precursors of steroid hormones but are also important components of cell membranes. Steroid hormones play a crucial role in both the initiation of egg production and the level of egg production [38,39]. Meanwhile, cell adhesion plays an important role in the selection and development of poultry follicles [40]. Our results support that steroid hormones and cell adhesion determines the laying performance of geese.

## 5. Conclusions

In conclusion, we found that the liver fat content of the high-, low-, and no-egg production group decreased in turn, while the abdominal fat weight increased in turn by phenotypic measurement. For transcriptional regulation, the number of DEGs in liver and abdominal fat with different laying performance was greater than that in ovarian stroma. Through functional analysis, the KEGG pathways related to lipid metabolism were enriched in all pairwise comparisons of abdominal fat and liver. However, some KEGG pathways related to inflammation and the circadian rhythm pathway were enriched by DEGs only in abdominal fat and the liver, respectively. The DEGs in ovarian stroma enriched some KEGG pathways related to ovarian steroidogenesis and cell adhesion.

## Figures and Tables

**Figure 1 animals-12-01775-f001:**
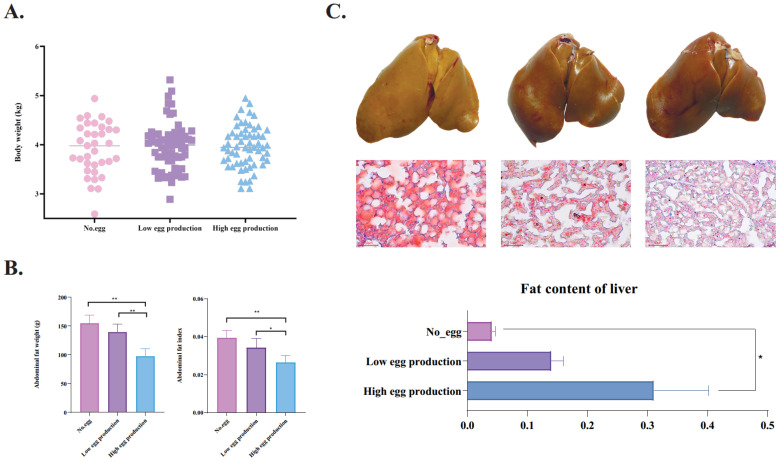
The lipid deposition patterns of geese with different laying performance. The body weight (**A**) and abdominal fat weight (index) (**B**) of different laying performance geese. (**C**) From left to right, the liver morphology and oil red staining sections of the high-egg production group, low-egg production group and no-egg group were in turn. Meanwhile, the OD values of oil red staining sections were counted. Data are shown as the mean ± standard deviation (SD). * means *p* < 0.05, and ** means *p* < 0.01.

**Figure 2 animals-12-01775-f002:**
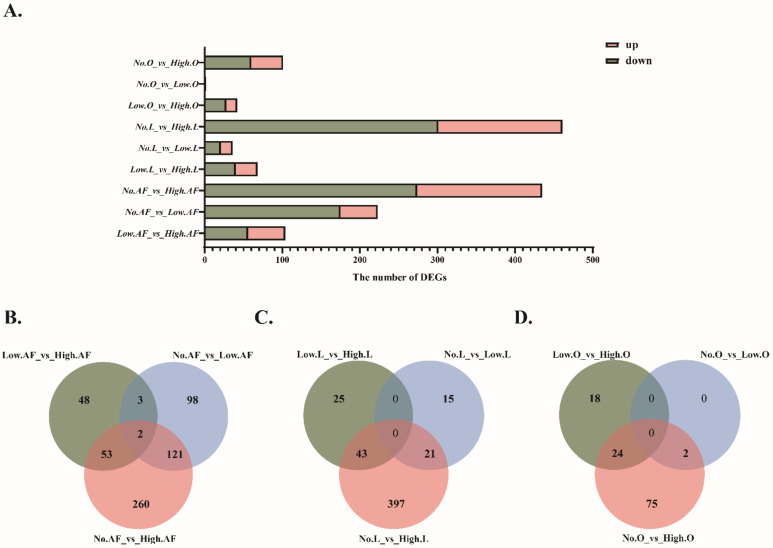
The number of DEGs in different groups. (**A**) Histogram of the number of DEGs in different groups. Venn diagram of the common DEGs between three pairwise comparisons in abdominal fat (**B**), liver (**C**), and ovarian stroma (**D**). “AF” means abdominal fat, “L” means liver, “O” means ovarian stroma.

**Figure 3 animals-12-01775-f003:**
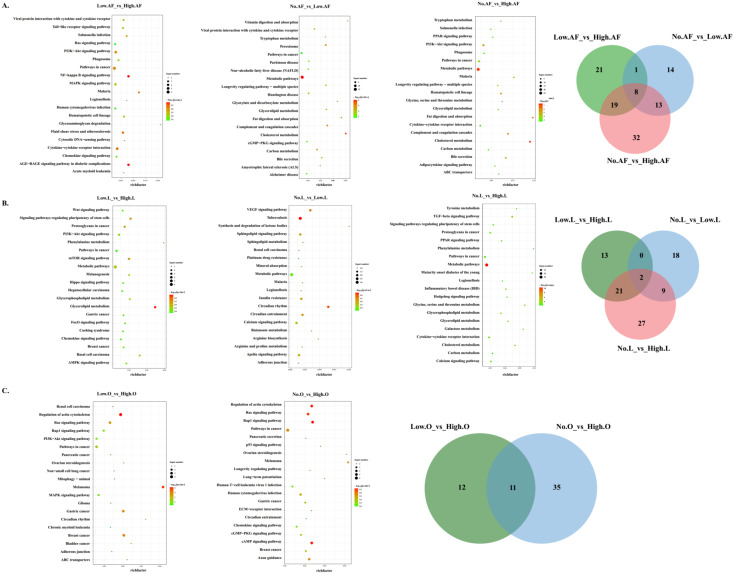
Functional annotation of the DEGs between different performances of geese in abdominal fat (**A**), liver (**B**), and ovarian stroma (**C**). “AF” means abdominal fat, “L” means liver, “O” means ovarian stroma.

**Figure 4 animals-12-01775-f004:**
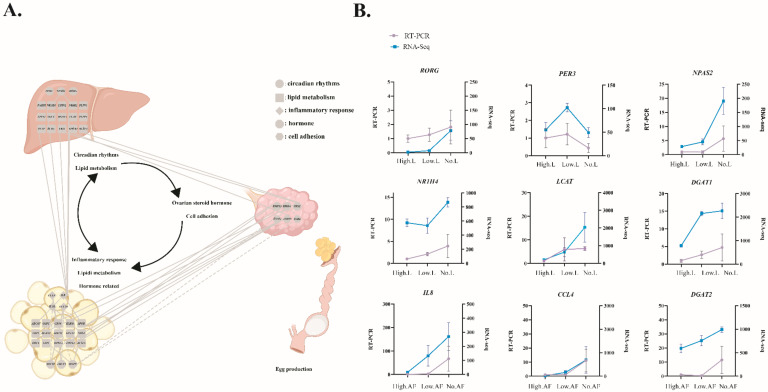
The network of liver, abdominal fat, and ovarian stroma regulating laying performance in geese and the expression validation of hub genes. The network of L–AF–O (liver, abdominal fat, ovarian stroma) regulating laying performance in geese (**A**). qRT-PCR validation of expression of the main DEGs involving the network of egg production performance regulation by liver, abdominal fat, and ovarian stroma (**B**). The results of qRT-PCR and RNA-Seq were expressed as the mean ± SD (n = 3, three biological replications at each group). “AF” means abdominal fat, “L” means liver.

**Table 1 animals-12-01775-t001:** Primers used for qRT-PCR in this study.

Primer Name	Sequence (5′–3′)	Product Length (bp)
*GAPDH-F*	GCTGATGCTCCCATGTTCGTGAT	86
*GAPDH-R*	GTGGTGCAAGAGGCATTGCTGAC	
*β-* *A* *CTIN* *-F*	CAACGAGCGGTTCAGGTGT	92
*β-* *A* *CTIN* *-R*	TGGAGTTGAAGGTGGTCTCGT	
*RORG-F*	TGTGCCAGAACGACCAGAT	102
*RORG-R*	AGAGGACGGTCCGGTTGT	
*PER3-F*	GAGCAGTGCCTTTGTTGGGT	276
*PER3-R*	TCAGAGGGCTTGTTCGGACT	
*NPAS2-F*	TCACAGAGCACCACCGATTA	148
*NPAS2-R*	ATAGCAACACGACTTCCCCT	
*NR1H4-F*	GCCTCAGATTTCATCGCCAC	228
*NR1H4-R*	GCTTTGTCACCACAGACCACG	
*LCAT-F*	CAGCGTGTCTTCCTCATTGC	187
*LCAT-R*	ACATAAGTGGGATGCCCTGAT	
*DGAT1-F*	GCCTACCCCGACAACCTCAC	180
*DGAT1-R*	CACCATCCACTGCTGGATCA	
*IL8-F*	CCTGGTAAGGATGGGAAACG	168
*IL8-R*	GGGTCCAAGCACACCTCTCT	
*CCL4-F*	ATGAAGGTCTCTGTGGCTGC	119
*CCL4-R*	TCCCGTTGGATGTAGGTGAA	
*DGAT2-F*	ACCCACAATCTGCTGACCAC	239
*DGAT2-R*	GATAAGATGTAGTCTATGCTGTCGC	

## Data Availability

The original sequencing data for this study can be found in the Sequence Read Archive (https://www.ncbi.nlm.nih.gov/sra; accessed on 22 January 2022) at NCBI with the BioProject ID: PRJNA747742.

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
