# Peer review of "Comparative Transcriptome Analysis Provides Novel Insights into the Effect of Lipid Metabolism on Laying of Geese"

_animals, 2022, doi:10.3390/ani12141775_

Round 1

Reviewer 1 Report

I am grateful to have had the opportunity to review this manuscript. It is based on important and interesting research in poultry science, egg production and lipid transport. 

I do have some comments that I hope the authors will find helpful. The most important being that the methods section needs to be expanded so that the reader can understand what was done. How many animals were sampled of which treatments and at what developmental stages.

1) One glaring concern of this manuscript is the grammar. I suggest that the authors seek out some grammatical editing. There are errors throughout this manuscript that make it difficult to read and, in some cases, make the exact meaning of statements unclear.

2)L-69: please clarify the type of geese used (strain and supplier). 

3) Section 2.1. Please expand on this section. How many geese were in the no eggs laid group? How many birds in the other groups? How birds were selected for slaughter is very confusing. Please expand with specifics. All birds selected for slaughter were between x and y kilograms (for example). And, what do you mean by "consistent physiological state"?

4)Clarify how liver was removed and put onto a tissue mold (L77) and then collected and placed in liquid nitrogen (L78-79)

5)L77 change optimal cutting temperature to "optimal cutting temperature compound"

6)Section 2.5 How were statistical comparison performed? T-test, ANOVA, etc

7) Figure 1 (and other figures) - In the figure caption, please indicate what data is shown and the meaning of the symbols used in the figure (ex. is the data LSMeans? Are the error bars SEM or SD? What is the P value cut off for * and **)

8)Figure 4. Here you say n=3, three biological replications at each developmental stage. This is the first mention n=3 or that there are samples from different biological stages. This information needs to be fully explained in the methods section

9) Supplemental tables are mentioned, but I don't have access to them.

10)Results section should include the statistical results. What were the significant findings and to what degree were they significant

Author Response

Response to reviewer1

I am grateful to have had the opportunity to review this manuscript. It is based on important and interesting research in poultry science, egg production and lipid transport. 

I do have some comments that I hope the authors will find helpful. The most important being that the methods section needs to be expanded so that the reader can understand what was done. How many animals were sampled of which treatments and at what developmental stages.

Reply:Thanks for your suggestion. We expanded the section of materials and methods, including the number of samples and their specific information.

1) One glaring concern of this manuscript is the grammar. I suggest that the authors seek out some grammatical editing. There are errors throughout this manuscript that make it difficult to read and, in some cases, make the exact meaning of statements unclear.

Reply:Aa you suggested, we have checked and corrected the grammar of the manuscript.

2)L-69: please clarify the type of geese used (strain and supplier). 

Reply:Done as you suggested.

3) Section 2.1. Please expand on this section. How many geese were in the no eggs laid group? How many birds in the other groups? How birds were selected for slaughter is very confusing. Please expand with specifics. All birds selected for slaughter were between x and y kilograms (for example). And, what do you mean by "consistent physiological state"?

Reply:Thanks for your suggestion. The number of geese in each group and selected geese has been added in the materials and methods section. The "consistent physiological state" means none of these selected individuals is in a molting state and all of them shows good mental state.

4)Clarify how liver was removed and put onto a tissue mold (L77) and then collected and placed in liquid nitrogen (L78-79)

Reply:The liver that put onto the module is the tissue used for oil red O staining. The liver that placed in liquid nitrogen is the tissue for RNA-seq. The tissues mentioned in these two parts are partial liver tissues, and presentation forms have been modified in the manuscript.

5)L77 change optimal cutting temperature to "optimal cutting temperature compound"

Reply:Done as requested.

6)Section 2.5 How were statistical comparison performed? T-test, ANOVA, etc

Reply:One-way ANOVA was used for statistical analysis in this study. The relevant content has been added in lines 126-127 of the manuscript.

7) Figure 1 (and other figures) - In the figure caption, please indicate what data is shown and the meaning of the symbols used in the figure (ex. is the data LSMeans? Are the error bars SEM or SD? What is the P value cut off for * and **)

Reply:Thanks for your suggestion. Data was shown as the mean ± standard error of mean (SEM). * means p<0.05, and **means p<0.01. These descriptions have been added to the figure caption.

8)Figure 4. Here you say n=3, three biological replications at each developmental stage. This is the first mention n=3 or that there are samples from different biological stages. This information needs to be fully explained in the methods section

Reply: Done as requested in line 94.

9) Supplemental tables are mentioned, but I don't have access to them.

Reply:Thank you for your suggestion. I have submitted supplemental table1 and supplemental table2.

10)Results section should include the statistical results. What were the significant findings and to what degree were they significant

Reply:Done as requested.

Reviewer 2 Report

Authors performed the transcriptome analysis to found genes to related to egg production and fat weight in geese. These fundamental data could contribute to geese breeding and industry. However, authors must modify many parts including to figure graphic and discussion before considering for publication.

Major points

1.       The reviewer cannot see the words in almost figures. They should modify the character size in figures and image quality. They should change the color of high egg production group in Figure 1A, since it is hard to understand for the reviewer. And also, they should change the words of Y axis in Figure 2A, since the reviewer cannot understand the meaning.

2.       Where is the supplemental tables?? The reviewer cannot find the tables.

3.       Authors indicated CCL4 and HTR3A as the important genes. Please show the log2 fold change or P value. And also they investigated gene expression of CCL4 using PCR analysis but not HTR3A. They should perform PCR analysis to investigate HTR3A if they consider HTR3A is one of the important gene to associate with egg production or fat metabolism.

4.       Authors performed the pathway analysis using KEGG pathway. The analysis found some important pathways to associate with egg production such as lipid metabolism, circadian rhythm, and hormone. In important pathways, authors should show the gene list and log2 fold change of the pathway as Tables.

5.       In discussion section, they use the pathways which is found in transcriptome to consider some possibilities. However, in important pathways, the changes of some important genes should be used for discussion. Authors should re-construct discussion section using the data of genes.

6.       In discussion, they state CCL4 and HTR3A relates to goose reproduction performance, using previous report (P8 L17-19). How these genes affect goose reproduction?? This is important point, so please state the detail mechanism.

7.       In introduction, they state that goose have stronger capacity for lipid metabolism in liver (P2 L49-50). Which pathways (lipid synthesis or both lipid synthesis and fatty acid oxidation) or which genes are activated or induced in goose? Please state in detail.

8.       Regarding to Figure legends, it is too simple to understand the figures and experiments. Authors should state more information in figure legends so that it would be easy to understand for readers.

Minor point

9.       What statistical analysis was carried out in this study?? One-way ANOVA?? Please indicate the information in Materials and Methods.

10.    The reviewer thinks the word of “Previously” could remove in P3 L109.

Author Response

Response to reviewer2

Authors performed the transcriptome analysis to found genes to related to egg production and fat weight in geese. These fundamental data could contribute to geese breeding and industry. However, authors must modify many parts including to figure graphic and discussion before considering for publication.

Major points

  1. The reviewer cannot see the words in almost figures. They should modify the character size in figures and image quality. They should change the color of high egg production group in Figure 1A, since it is hard to understand for the reviewer. And also, they should change the words of Y axis in Figure 2A, since the reviewer cannot understand the meaning.

Reply:Thanks for your suggestion. The Word software may compress the quality of the figures. Therefore, the PDF version of these figures is provided in the attachment. We hope that the fonts in these pictures can be seen. Meanwhile, we have modified the color of Figure 1A and added the ordinate legend of Figure 2A.

  1. Where is the supplemental tables?? The reviewer cannot find the tables.

Reply:Thank you for your suggestion. I have uploaded supplemental table1 and supplemental table2 in the revised version.

  1. Authors indicated CCL4 and HTRA3 as the important genes. Please show the log2 fold change or P value. And also they investigated gene expression of CCL4 using PCR analysis but not HTRA3. They should perform PCR analysis to investigate HTRA3 if they consider HTRA3 is one of the important gene to associate with egg production or fat metabolism. 6. In discussion, they state CCL4 and HTRA3 relates to goose reproduction performance, using previous report (P8 L17-19). How these genes affect goose reproduction?? This is important point, so please state the detail mechanism.

Reply:Thanks for your suggestion. The purpose of this study is to explain the potential pathways of lipids involved in reproductive regulation. The important DEGs in liver alone may not be the focus of this study. In order to make readers more clearly understand the main idea of this study, we have removed the relevant parts of the manuscript in the revised version.

  1. Authors performed the pathway analysis using KEGG pathway. The analysis found some important pathways to associate with egg production such as lipid metabolism, circadian rhythm, and hormone. In important pathways, authors should show the gene list and log2 fold change of the pathway as Tables.

Reply:Thank you for your suggestion. The list of DEGs that enriched in KEGG pathway is shown in Table 2. At present, supplemental table2 has been submitted.

  1. In discussion section, they use the pathways which is found in transcriptome to consider some possibilities. However, in important pathways, the changes of some important genes should be used for discussion. Authors should re-construct discussion section using the data of genes.

Reply:As you suggested, we re-construct the structure of the discussion section and focus on the key DEGs in key pathways.

  1. In introduction, they state that goose have stronger capacity for lipid metabolism in liver (P2 L49-50). Which pathways (lipid synthesis or both lipid synthesis and fatty acid oxidation) or which genes are activated or induced in goose? Please state in detail.

Reply:Phenotypically, when geese were overfed on the high-energy diet, liver volume increased 5-10 times within 2 weeks, and liver returned to its original state when the geese returned to the regular diet for 20 days. Therefore, geese are commonly used to produce fatty liver, a phenomenon that does not exist in land poultry. Some scholars have pointed out that compared with adipose tissue, the difference in lipid metabolism caused by overfeeding is more obvious in the liver. The liver involves pathways such as lipid metabolism and immune response, while the adipose tissue involves pathways such as protein binding and gluconeogenesis. We have added these contents in lines 49-53 of the manuscript.

  1. Regarding to Figure legends, it is too simple to understand the figures and experiments. Authors should state more information in figure legends so that it would be easy to understand for readers.

Reply:Thanks for your suggestion. We have added more complete legend content in the revised version to make it easier to understand.

Minor point

  1. What statistical analysis was carried out in this study?? One-way ANOVA?? Please indicate the information in Materials and Methods.

Reply:One-way ANOVA was used for statistical analysis in this study. The relevant content has been added in lines 126-127 of the manuscript.

  1. The reviewer thinks the word of “Previously” could remove in P3 L109.

Reply:Done as requested.

Round 2

Reviewer 1 Report

Thank you for submitting your changes and answers to the reviewers comments and questions. After reviewing this manuscript again, I strongly suggest having an additional person review this manuscript to assist in grammar/editing.

Author Response

Thanks again for your suggestion. This manuscript has been checked by native English-speaking colleague.

Reviewer 2 Report

The revised manuscript has improved adequately. I have still some requests for the illustration of Figures.

1.       I cannot see the provided clear Figures (PDF format).  Next time, please provide us the clear Figures.

In revised manuscript, I noticed that the color of groups was differed in figures. For example, the color of No egg group is blue in Figure 1A, but it is green in Figure 1B. And also, the order of groups in Figure 1A and 1B is differs. These points leads to misunderstand for readers, so please modify these points.

2.       The Y axis of Figure 2A is still hard to understand. Please change the word or add the explanation in Figure legends.

Author Response

Thanks again for your suggestion. We have modified the color and order of Fig1. 

The explanation of the words mentioned in Fig.2 was added.